# The Influence of Annealing on the Microstructural and Textural Evolution of Cold-Rolled Er Metal

**DOI:** 10.3390/ma15248848

**Published:** 2022-12-11

**Authors:** Shiying Chen, Yixuan Wang, Xiaowei Zhang, Jinying Li, Zongan Li, Wensheng Yang, Daogao Wu, Zhiqiang Wang, Dehong Chen, Ning Mao

**Affiliations:** 1National Engineering Research Center for Rare Earth, GRINM Group Co., Ltd., Beijing 100088, China; 2GRIREM Advanced Materials Co., Ltd., Beijing 100088, China; 3General Research Institute for Nonferrous Metals, Beijing 100088, China; 4School of Materials Science and Engineering, South China University of Technology, Guangzhou 510640, China

**Keywords:** erbium target, annealing, microstructure, texture, grain size

## Abstract

The microstructural and textural evolution of 60% cold-rolling-deformation Er metal (purity ≥ 99.7%) during annealing were investigated by electron-backscattered diffraction (EBSD) and X-ray diffraction (XRD). The research results showed that the texture of the (0001) plane orientation was strengthened, but there was no apparent enhancement of the (01$\bar{1}$0) and ($\bar{1}$2$\bar{1}$0) plane orientations with increasing the annealing temperature. The recrystallization frequency and grain sizes gradually stabilized after the annealing duration of more than 1 h at 740 °C; the annealing duration and the recrystallization frequency were fitted to the equation: $y=1\;-\;\exp\;(-0{.3269x}^{0.2506})$. HAGBs were predominant, and the distribution of grain sizes was the most uniform after annealing at 740 °C × 1 h, which was the optimal annealing process of the Er metal with 60% cold-rolling deformation. However, the recrystallization was transferred to the substructure due to grain boundary migration and twining under an excessive annealing temperature and duration.

## 1. Introduction

Rare-earth thin-film material is a crucial material for the national defense and emerging strategic industry [1,2,3]. Examples include Er film for T-target materials in the nuclear industry, which have the advantages of better thermal stability and a high helium (3 He)-release threshold [4,5,6,7]. Er_2_O_3_ thin film has been applied in new-generation electronic information with high-dielectric-constant material, better thermal stability, and a wide bandgap [8,9,10,11]. A high sputtering efficiency and stable films can be easily achieved if the targets have features with refined grains, particular orientations, and homogenized microstructures. So, the uniformity of the microstructure, grain size, and orientation distribution is of great practical significance to the Er target, as the source material of thin Er_2_O_3_ and Er films.

There are problems with melt-casting metal targets, such as them having a coarse grain size, an uneven microstructure distribution, etc. Plastic deformation and annealing are usually applied to obtain a refined grain size, homogeneous microstructure, and particular crystal orientation [12,13,14,15,16,17]. The plastic deformation of rare earth metals is generally carried out by cold-rolling to avoid severe oxidation and dynamic recrystallization. However, Er has a hexagonal close-packed (HCP) crystal structure, which is easy to crack due to work hardening. The mechanism of microplastic deformation is complex, and deformed grains grow abnormally during heat treatment.

At present, there is a lot of research on the cold plastic deformation and heat treatment of dense hexagonal-structure metals, such as titanium [18,19,20], magnesium [21,22], and magnesium alloys [23,24,25], but there have been fewer studies on the plastic deformation and heat-treatment process of rare earth metals. S. Taskaev [26,27,28,29] refined the coarse grains of terbium, dysprosium, and gadolinium by optimizing cold plastic deformation and annealing to improve the materials’ magnetic and magneto-thermal properties. Huang Pei [30] obtained a high-quality scandium target material with a fine grain and uniform microstructure by adjusting the total forging ratio and optimizing the heat-treatment process. Wang Shuang [31] obtained Er targets with uniform microstructures and refined grains by hot-rolling and heat treatment. Still, the hot-rolling process resulted in severe oxidation of the Er and the abnormal growth of some grains. Previous studies have obtained the desired microstructure, texture evolution, and deformation mechanism during cold-rolling [32]. However, there is a lack of systematic and in-depth research on the microstructure and texture evolution of the cold-rolling of Er targets during heat treatment.

In this study, Er metal was rolled to 60% deformation by combining the cold-rolling and intermediate annealing processes. The effect of the annealing temperature and duration on the microstructure and texture of the Er target was investigated during annealing. The evolution of the Er target in terms of the microstructure, texture during annealing, and the optimized annealing process of the 60% cold-rolling deformation were obtained.

## 2. Materials and Methods

The initial Er metal (99.7 wt.%) sample of 50 mm × 30 mm × 12 mm was deformed to a 60% cold-rolling deformation (due to the excellent grain crushing and refinement under a 60% cold-rolling deformation) by a two-high mill with a diameter of 500 mm, and the reduction in each pass was 3% of the thickness. Stress-relief annealing was performed at 520 °C × 1 h for each 10% reduction along the thickness during the cold-rolling process. Then, the square [10 mm (length) × 10 mm (width) × 10 mm (thickness)] samples taken from the center of the 60% cold-rolled Er metal were homogenously annealed in a vacuum heat-treatment furnace. The annealing temperatures were 460 °C, 500 °C, 540 °C, 620 °C, 660 °C, 700 °C, 740 °C, 780 °C, and 820 °C, and the annealing duration was 1 h. Furthermore, the annealing duration experiments were performed at 740 °C, and the annealing durations were 0.5 h, 1 h, 1.5 h, 2 h, and 2.5 h, respectively. A schematic diagram of the rolling and sampling is proven in Figure 1.

The annealed samples were polished with SiC sandpaper of 200 mesh, 600 mesh, 1500 mesh, 3000 mesh, and 5000 mesh. The samples were mechanically polished with a 0.5 μm polishing agent until there were no apparent scratches under an optical microscope (OM). Furthermore, samples for EBSD measurement were further polished with argon ions after mechanical polishing. The microstructure and texture in the rolling direction (RD) and normal direction (ND) planes at the mid-width of the sample were observed by EBSD. EBSD analysis was performed with a ZEISS-Sigma500 scanning electron microscope equipped with an Oxford-symmetry accessory. The crystal orientation in the RD–ND plane was analyzed by a Philips PANalytical XRD unit (PW3373, Almelo, The Netherlands) with Cu-Kα radiation.

## 3. Results

### 3.1. Sixty PercentCold-Rolling Deformation State of the Er Metal

Our previous studies showed that the initial grain sizes were large than 200 mm [32]. Figure 2 shows the inverse pole figure (IPF), pole figure, and grain size distribution of the 60% cold-rolled Er metal. Figure 2a displays that the coarse grains were broken and refined entirely, and a broken deformed structure was dominant. Some dynamically recrystallized grains appeared inside the grains in the circle of Figure 2a, indicating that the grains were crushed completely. Figure 2b declares that the main orientations included (0001), ($\bar{1}$2$\bar{1}$0), and (01$\bar{1}$0), and the texture intensity was less than 10, revealing no apparent texture phenomenon. The grain size distribution was mainly concentrated in the range of 0–10 μm, as shown in Figure 2c, presenting the grain size as being fine and uniform distribution.

### 3.2. Sixy Percent Cold-Rolling Deformation State of the Er Metal

Figure 3 is the IPF of the Er metal with various annealing temperatures. As shown in Figure 3a, when the annealing temperature rose to 460 °C, the many deformed structures and the small amount of recrystallization in the cracked grains indicated an initial recrystallization state. It can be observed from Figure 3a–c that when the annealing temperature increased from 460 °C to 540 °C, the deformation of the structure was gradually reduced, and many recrystallized grains appeared. However, the relatively low annealing temperature provided insufficient heat energy for the recrystallized grains to grow adequately. Therefore, there were still some deformation structures, and the distribution of the recrystallized grains was not uniform. It can be seen in Figure 3d–f that when the annealing temperature increased from 620 °C to 700 °C, the recrystallization was further improved, and the deformation structures were replaced entirely by equiaxed crystals. However, there were still a large number of refined grains because the heat treatment temperature was relatively low enough to not provide enough heat energy to make the grains with a lower internal stress storage energy position fully grown. As shown in Figure 3g, the grain distribution was uniform, and the deformation area was replaced by refined equiaxed grains, indicating that the recrystallization finished when the annealing temperature increased to 740 °C. When the annealing temperature rose from 780 °C to 820 °C, there were some abnormal coarsened grains because the high-energy grain boundaries migrated and swallowed refined grains, and some annealing twins are shown in the circle of Figure 3h, indicating that the uniformity of grains size decreased.

Figure 4 is the grain size distribution with various annealing temperatures, and Figure 5 presents the relationship curve between the average grain size and the temperature. As shown in Figure 4a–c, the grain size was mainly distributed in the range of 0–10 μm, but a few grains were more extensive than 70 μm, indicating that the grain size distribution was not uniform when the temperature was less than 540 °C. Figure 4d–f displays that the distribution of grain sizes was even more than that at 540 °C, and the grain size was mainly distributed in the range of 0–40 μm when the temperature increased from 620 °C to 700 °C. As shown in Figure 4g, the grain size distribution was the most uniform and was concentrated in the range of 0–30 μm when the temperature increased to 740 °C. However, some grain sizes were more extensive than 50 μm, and the uniformity of the grain sizes decreased as the annealing temperature increased from 780 °C to 820 °C (Figure 4h,i). Figure 5 indicates that the average grain size first increased slowly and then increased rapidly, and the average grain size growth tended to be stable when the annealing temperature was higher than 740 °C. So, the best heat-treatment temperature was 740 °C under the above annealing process.

Figure 6 shows the misorientation angle distributions for the 60% cold-rolled Er metal with various annealing temperatures. As shown in Figure 6a–c, when the annealing temperature was less than 540 °C, the misorientation was distributed randomly; there were a lot of low-angle grain boundaries (LAGBs, θ < 15°, more than 70.5%), but fewer high-angle grain boundaries (HAGBs, θ > 15°, less than 29.5%). After the annealing temperature increased from 540 °C to 740 °C (Figure 6c–g), the frequency of LAGBs decreased, and HAGBs increased from 27.16% to 72.32%. In addition, the misorientation was mainly distributed in the range of 15–40°. However, when the annealing temperature increased from 740 °C to 820 °C (Figure 6g–i), the HAGB frequency decreased from 72.32% to 50.73% due to grain boundary migration and twins. Figure 7 displays that the highest content of HAGBs was observed at 740 °C. The results of the misorientation angle distributions (Figure 6) and the curve of high-angle grain boundary content with annealing temperature (Figure 7) indicated that recrystallization was completed at 740 °C.

Figure 8 shows the grain characteristics distribution of the 60% cold-rolled Er metal with various annealing temperatures, and Table 1 presents the proportion of the recrystallization area and deformation area accordingly. The recrystallization area ratio was less than 7%, and the deformation area was around 90% at a lower temperature (Figure 8a–c). The recrystallization area proportion increased from 6.95% to 95.24%, and the ratio of deformation area decreased to 0.4% when the temperature increased from 540 °C to 740 °C (Figure 8c–g). However, the proportion of the recrystallization area was rapidly reduced after the annealing temperature increased to 820 °C, which could be because of the many substructures generated by the grain boundary migration and twins at a high temperature. Therefore, it was concluded that the recrystallization was completed at 740 °C, which was consistent with the result shown in Figure 3.

Figure 9 is the XRD map of the 60% cold-rolled Er with various annealing temperatures. It shows that the crystal orientations of Er metal mainly included the (01$\bar{1}$0), (0001), ($\bar{1}$2$\bar{1}$0), (10$\bar{1}$1), and (11$\bar{2}$2) planes in the present experimental annealing temperature. The intensity of the oriented peaks changed with the increase in the annealing temperature but did not directly reflect the changing trend of the crystal orientation relative to the initial Er sample. Therefore, the intensity ratio of the two characteristic peaks, R, with a preferred orientation, the intensity ratio of the two characteristic peaks, R0, with no preferred orientation, and the orientation index, P, were introduced, calculated as follows:(1)$$P=\frac{R}{R_{0}}=\frac{I_{{(H}_{1}K_{1}L_{1})}/I_{{(H}_{1}^{\prime }K_{1}^{\prime }L_{1}^{\prime })}}{I_{{(H}_{0}K_{0}L_{0})}/I_{{(H}_{0}^{\prime }K_{0}^{\prime }L_{0}^{\prime })}}$$
where $I_{{(H}_{1}K_{1}L_{1})}$ and $I_{{(H}_{1}^{\prime }K_{1}^{\prime }L_{1}^{\prime })}$ are the diffraction intensity of the (H_1_K_1_L_1_) and (H’_1_K’_1_L’_1_) planes with a preferred orientation, and $I_{{(H}_{0}K_{0}L_{0})}$ and $I_{{(H}_{0}^{\prime }K_{0}^{\prime }L_{0}^{\prime })}$ are the diffraction intensity of the (H_0_K_0_L_0_) and (H’_0_K’_0_L’_0_) planes with no preferred orientation, respectively. There is no preferred orientation when the orientation index P equals 1. On the contrary, there is the preferred orientation when P deviates from 1, and the greater the deviation, the more pronounced the preferred orientation [33].

In this research, (11$\bar{2}$2) was the reference plane, and I_0_ was the orientation intensity value of (11$\bar{2}$2) under various annealing temperatures. P1, P2, P3, and P4 are the orientation indices of the (01$\bar{1}$0), (0001), (10$\bar{1}$1), and ($\bar{1}$2$\bar{1}$0) planes, respectively. The calculation results are shown in Table 2, and Figure 10 is the curve of the orientation index with annealing temperature. It displays that the orientation index of the (0001) plane was more than 1.49, and there was a significant deviation of the orientation index from 1. However, the orientation indices of (01$\bar{1}$0), (10$\bar{1}$1), and ($\bar{1}$2$\bar{1}$0) were between 0.75 and 1.3, which is around 1. Therefore, it indicated that the preferred orientation of the (0001) plane had a further enhancement trend, but the (01$\bar{1}$0), (10$\bar{1}$1), and ($\bar{1}$2$\bar{1}$0) planes remained unchanged with increasing the annealing temperature.

Figure 11 shows the (0001), (11$\bar{2}$0), and (10$\bar{1}$0) pole figures of the Er metal with 60% cold-rolling deformation annealed at various temperatures. It was observed that the texture intensity distribution of the (0001) plane was concentrated, presenting that there was an obviously preferred orientation. In contrast, the texture intensity distribution of (01$\bar{1}$0) and ($\bar{1}$2$\bar{1}$0) dispersed, indicating no preferred orientation with increasing the temperature. The maximum texture intensities increased from 9.55 to 19.14 as the annealing temperature increased from the initial cold-rolling state to 660 °C. The texture intensity was more than 15 after annealing, indicating a strongly preferred orientation in the (0001) plane during the annealing process. The results of the texture variation in Figure 11 were consistent with Figure 10.

### 3.3. Effect of Annealing Duration on the Microstructure of Er Metal

To further obtain the optimal annealing process, the effect of the annealing duration on the microstructure of the Er metal with 60% cold-rolling deformation was studied at 740 °C. Figure 12a displays that many refined recrystallized grains appeared inside the deformed structure. However, there were still a few deformed structures, indicating that the grain size was unevenly distributed and incompletely recrystallized. As shown in Figure 12b, the deformation structure was replaced by refined equiaxed grains, and the microstructure and grain size distribution were uniform when the annealing duration increased to 1 h, presenting that the recrystallization process finished. Figure 12c shows that the result of the recrystallization with an annealing duration of 1.5 h was similar to the annealing duration of 1 h, but the coarsening of a few grains was obvious. The dropped uniformity of the grain sizes could be due to the prolonged annealing duration. However, when the annealing duration further increased to more than 2 h (Figure 12d,e), the uniformity of the grain size distribution decreased, and some grains grew abnormally due to secondary recrystallization. So, it was concluded that the best heat treatment duration was 1 h under 740 °C.

Table 3 shows the recrystallization frequency of the Er metal with 60% cold deformation with various annealing durations at 740 °C. It displays that the recrystallization frequency reached 95.3% when the annealing duration increased to 1 h and then gradually stabilizes. Figure 13 shows the curve of the recrystallization frequency and annealing duration accordingly, and the relation of the curve fitting equation was calculated as follows:(2)$$y=1-\exp(-0{.3269x}^{0.2506})\;$$
where y is the recrystallization frequency and x is the annealing duration. The curve-fitting equation conformed to the Avrami equation:(3)$$\varphi_{R}=1-\exp(-{\mathrm{kt}}^{n})\;$$
where, φ_R_ is the recrystallization frequency, t is the annealing time, k is the Avrami constant, and n is the Avrami exponent. As usual, exponent n was between 1 and 4, but the curve-fitting equation exponent n = 0.2506 ≤ 1 due to selecting only five annealing durations; therefore, there was a significant deviation from the theoretical value. The Avrami recrystallization kinetics equation describes the relationship between the recrystallization frequency and annealing time, showing that the recrystallization frequency will be gradually stable after reaching the completed recrystallization time [34,35]. Therefore, it further indicated that the recrystallization was completed when the annealing time reached 1 h at 740 °C.

Figure 14a–e shows the grain size distribution of the Er metal with 60% cold-rolling deformation with various annealing durations at 740 °C. Figure 14a illustrates that the grain size distribution was mainly concentrated in the range of 0–30 μm. Still, a few grain sizes were more extensive than 100 μm, so the grain size distribution was not uniform after the annealing duration reached 0.5 h. The grain size was mainly distributed in the range of 0–20 μm, and the maximum grain size was not more than 40 μm when the annealing duration increased to 1 h (Figure 14b); thus, the grain size distribution was uniform. As shown in Figure 14c–e, the distribution of the grains was mainly concentrated in the range of 0–30 μm. Still, a few grains were more extensive than 60 μm, or even more than 80 μm, after the annealing duration increased to 1.5 h, 2 h, and 2.5 h; therefore, the uniformity of the grain size decreased significantly. Figure 14f shows the curve of the average grain size and annealing duration at 740 °C, and it displays that the average grain size increased rapidly from 3.68 μm to 9.03 μm when the annealing duration increased from 0 h to 1 h. The average grain size increased gently when the annealing duration increased from 1 h to 2.5 h. As a result, it indicates that the grain size distribution was the most uniform, and the recrystallization was completed at 740 °C × 1 h, coinciding with the results of Figure 12 and Figure 13.

Figure 15 shows the ideal microstructure evolution of the Er metal in the cold-rolling and annealing process. It indicates that the coarse, heterogeneous, and equiaxed grains were crushed into fine deformed structures, and then equiaxed refined grains were obtained by an optimal annealing process. In conclusion, the experiment result of the Er metal’s microstructure evolution was consistent with the ideal results during the cold-rolling and annealing process.

## 4. Conclusions

Er metal samples with 60% cold-rolling deformation were annealed with various annealing temperatures and durations, and the evolution of the microstructure and texture was investigated. The main conclusions of this work are summarized as follows:

The deformed structures of Er metal with 60% cold-rolling deformation were gradually replaced by homogeneous and refined equiaxial grains during the annealing process. The distribution of the grain sizes was the most uniform after annealing under the optimal annealing process of 740 °C × 1 h, with this sample showing the best recrystallization. However, the recrystallization was transferred to the substructure due to grain boundary migration and twins under the excessive annealing temperature and duration.

The recrystallization frequency and grain sizes gradually stabilized after the annealing duration was more than 1 h at 740 °C. The curve of the recrystallization frequency for the Er metal with 60% cold-rolling deformation and an annealing duration at 740 °C was fitted to the equation: $y=1-\exp\;(-0{.3269x}^{0.2506})$. Compared with the initial cold-rolling state, the orientation index of the (0001) plane was much larger than 1; thus, there was a tendency to further strengthen the preferred orientation of the (0001) plane. However, there were no apparent preferred orientations of the (01$\bar{1}$0), ($\bar{1}$2$\bar{1}$0), and (11$\bar{1}$0) plane orientations, indicating that the annealing contributed to the generation of the (0001) plane orientation.

## Figures and Tables

**Figure 1 materials-15-08848-f001:**
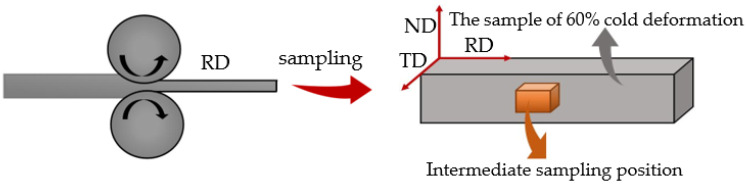
Cold-rolling and sampling.

**Figure 2 materials-15-08848-f002:**
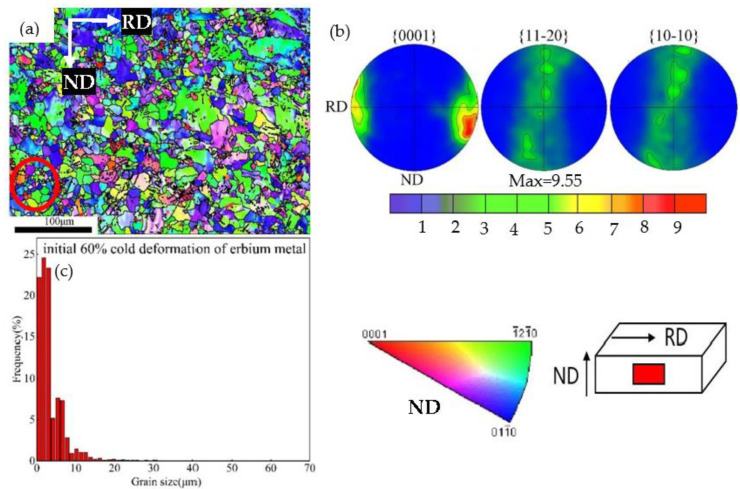
Sixty percent cold-rolling deformation of the Er metal: (**a**) inverse pole figure (IPF); (**b**) (0001), (11-20), and (10-10) pole figure; (**c**) grain size distribution.

**Figure 3 materials-15-08848-f003:**
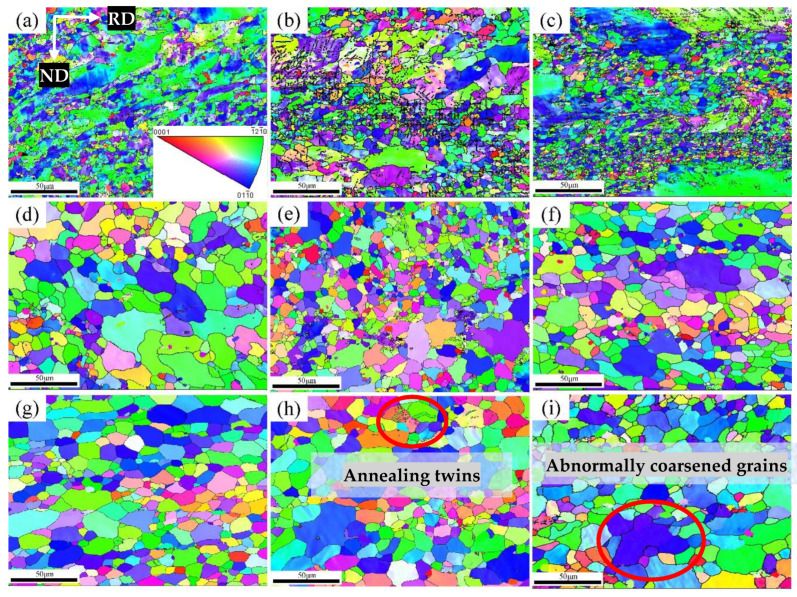
IPF of Er metal with 60% cold-rolling deformation with various annealing temperatures: (**a**) 460 °C; (**b**) 500 °C; (**c**) 540 °C; (**d**) 620 °C; (**e**) 660 °C; (**f**) 700 °C; (**g**) 740 °C; (**h**) 780 °C; (**i**) 820 °C.

**Figure 4 materials-15-08848-f004:**
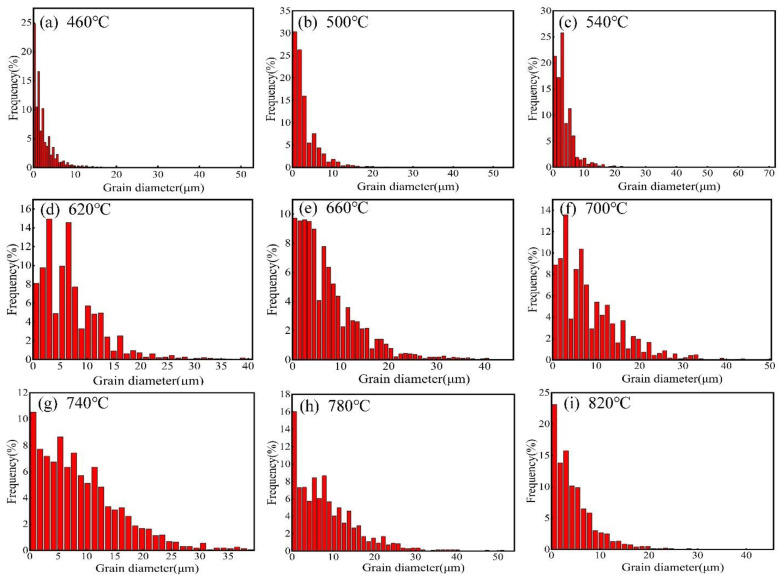
Grain size distribution of (**a**) 460 °C; (**b**) 500 °C; (**c**) 540 °C; (**d**)620 °C; (**e**) 660 °C; (**f**) 700 °C; (**g**) 740 °C; (**h**) 780 °C; and (**i**) 820 °C.

**Figure 5 materials-15-08848-f005:**
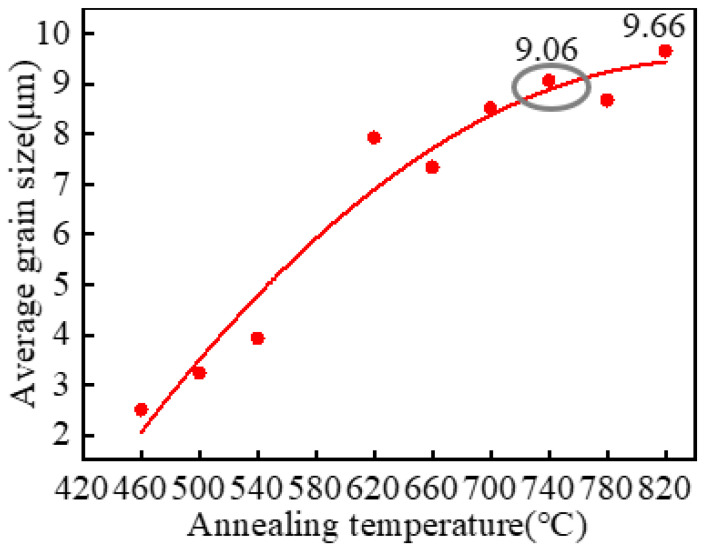
The relationship curve between the average grain size and the annealing temperature.

**Figure 6 materials-15-08848-f006:**
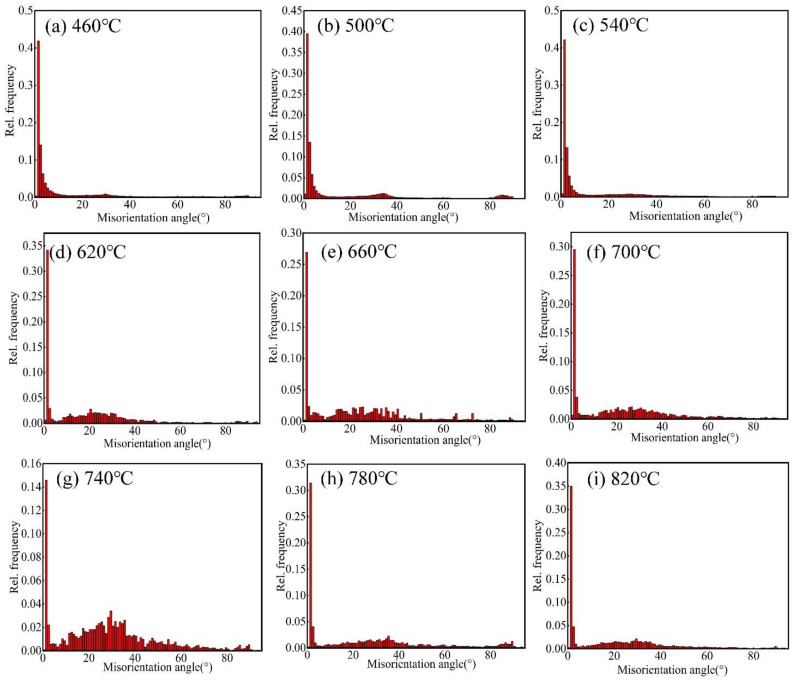
Misorientation angle distributions for the 60% cold-rolled Er metal with various annealing temperatures: (**a**) 460 °C; (**b**) 500 °C; (**c**) 540 °C; (**d**) 620 °C; (**e**) 660 °C; (**f**) 700 °C; (**g**) 740 °C; (**h**) 780 °C; and (**i**) 820 °C.

**Figure 7 materials-15-08848-f007:**
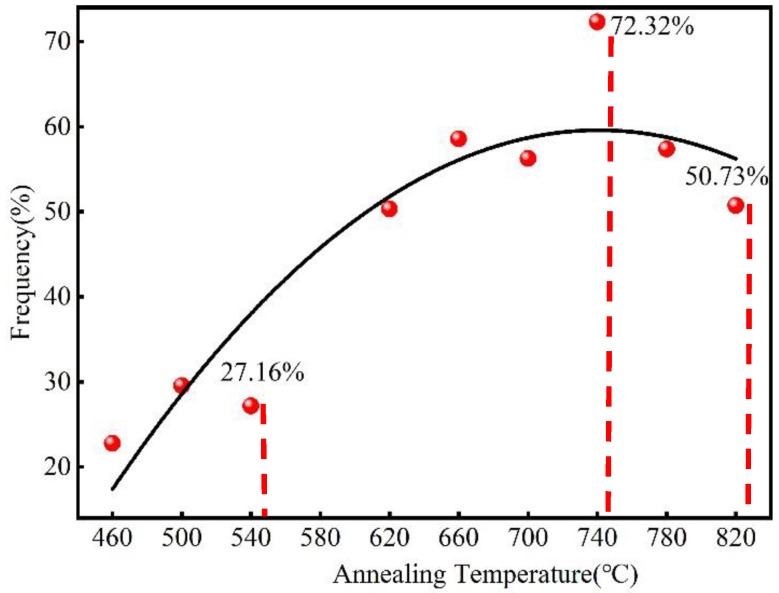
The curve of high-angle grain boundary content and annealing temperature.

**Figure 8 materials-15-08848-f008:**
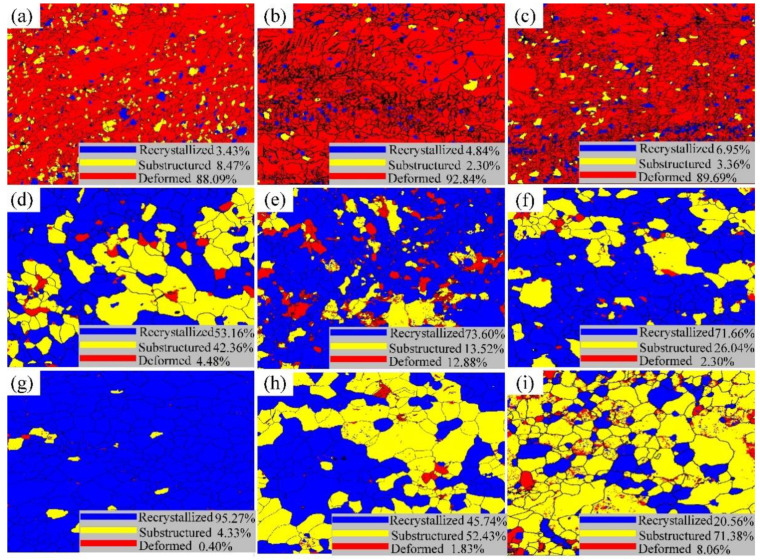
Grain characteristics distribution of 60% cold-rolled Er metal with various annealing temperatures: (**a**) 460 °C; (**b**) 500 °C; (**c**) 540 °C; (**d**) 620 °C; (**e**) 660 °C; (**f**) 700 °C; (**g**) 740 °C; (**h**) 780 °C; and (**i**) 820 °C.

**Figure 9 materials-15-08848-f009:**
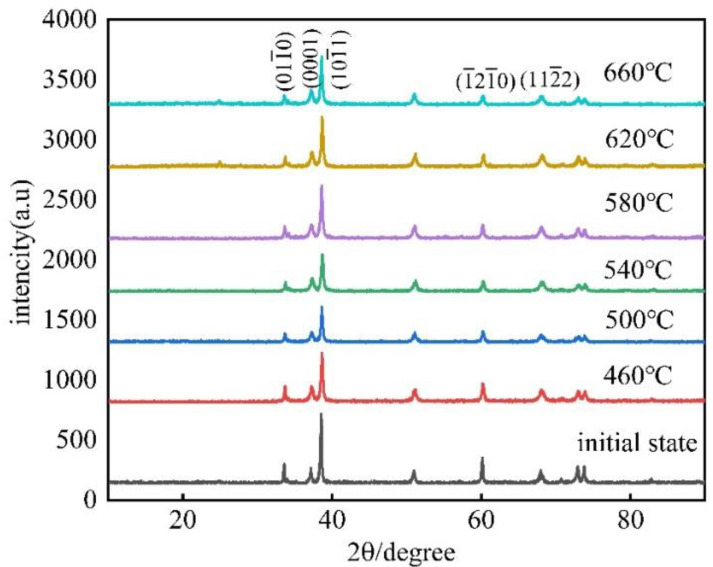
XRD map of 60% cold-rolled Er with various annealing temperatures.

**Figure 10 materials-15-08848-f010:**
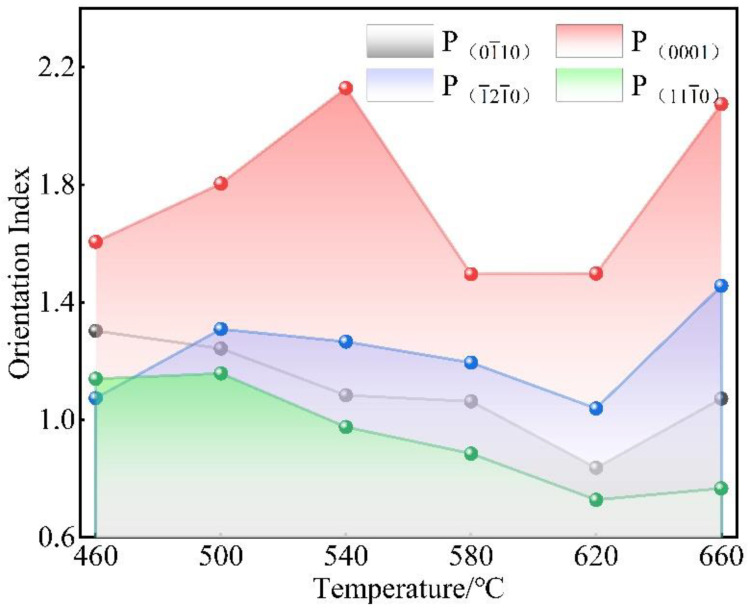
The curve between the orientation index and different annealing temperatures.

**Figure 11 materials-15-08848-f011:**
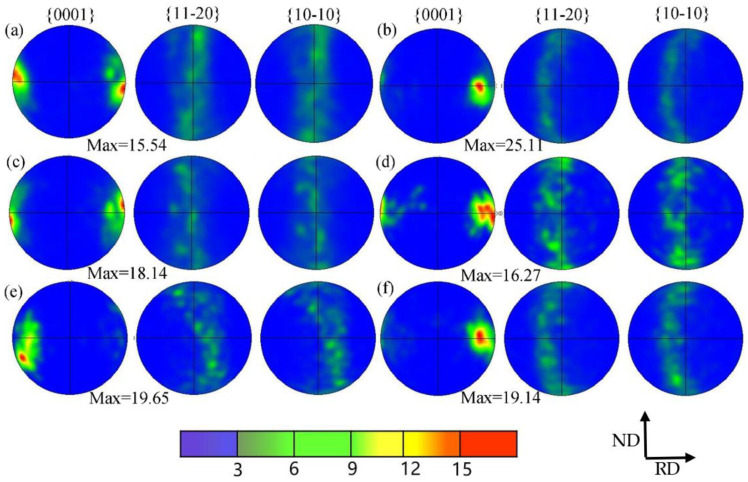
(0001), (11-20), and (10-10) pole figures in the Er metal with 60% cold-rolling deformation annealed at different temperatures: (**a**) 460 °C; (**b**) 500 °C; (**c**) 540 °C; (**d**) 580 °C; (**e**) 620 °C; and (**f**) 660 °C.

**Figure 12 materials-15-08848-f012:**
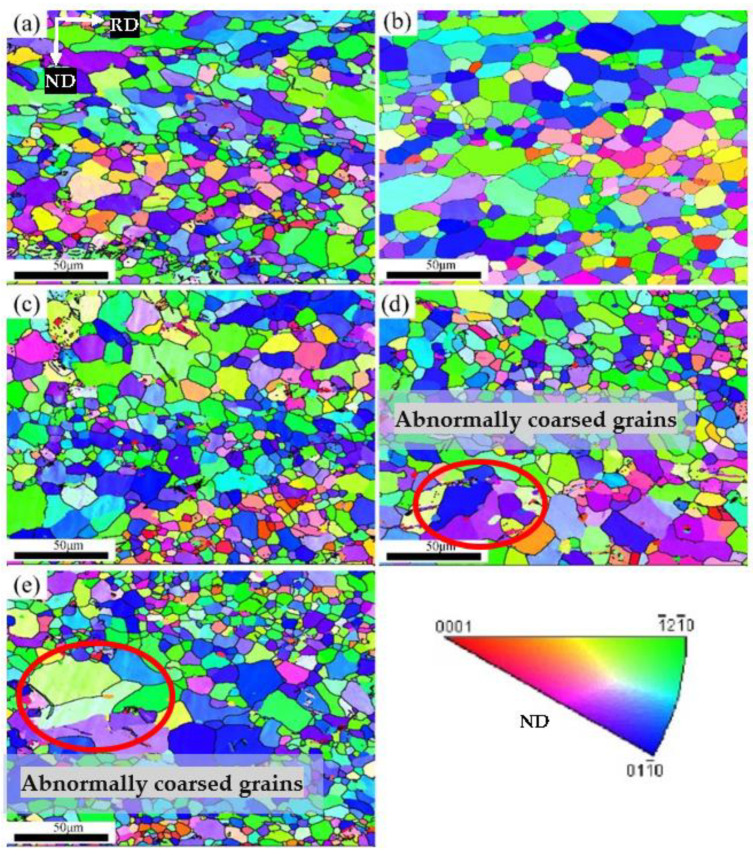
IPF of Er metal with 60% cold-rolling deformation with various annealing durations at 740 °C: (**a**) 0.5 h; (**b**) 1 h; (**c**) 1.5 h; (**d**) 2 h; and (**e**) 2.5 h.

**Figure 13 materials-15-08848-f013:**
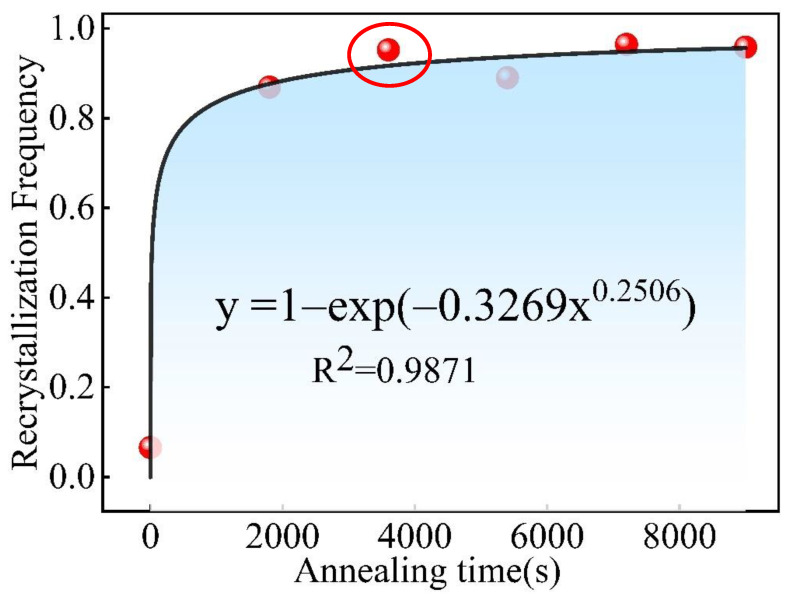
The curve of recrystallization frequency and annealing duration.

**Figure 14 materials-15-08848-f014:**
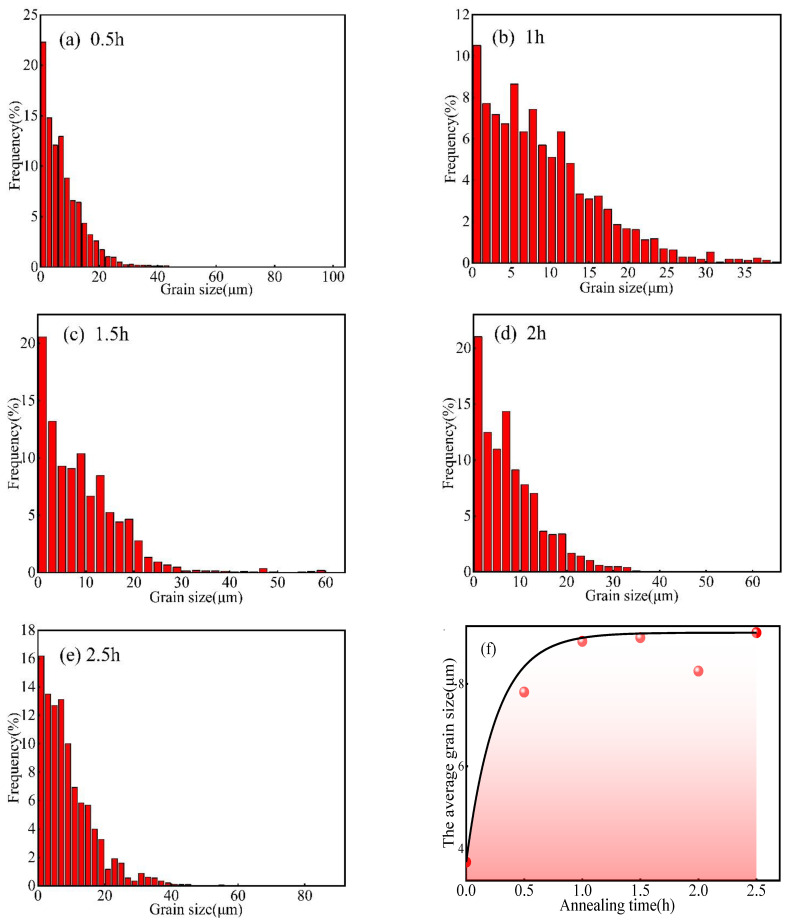
Grain size distribution of Er metal with 60% cold-rolling deformation with various annealing durations: (**a**) 0.5 h; (**b**) 1 h; (**c**) 1.5 h; (**d**) 2 h; and (**e**) 2.5 h. (**f**) The curve of the average grain size and annealing duration.

**Figure 15 materials-15-08848-f015:**
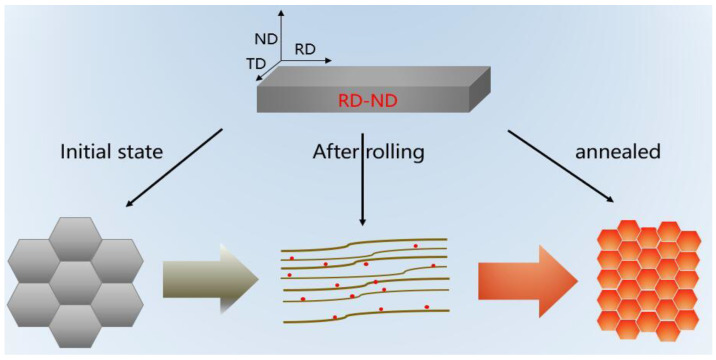
The ideal evolution of the Er metal’s microstructure in the cold-rolling and annealing process.

**Table 1 materials-15-08848-t001:** The proportion of recrystallization and deformation area of the 60% cold-rolled Er metal with various annealing temperatures.

Temperature (℃)	Recrystallized (%)	Substructure (%)	Deformed (%)
460	3.43	8.47	88.09
500	4.84	2.30	92.84
540	6.95	3.36	89.69
620	53.16	42.36	4.48
660	73.60	13.52	12.88
700	71.66	26.04	2.30
740	95.27	4.33	0.40
780	45.74	52.43	1.83
820	20.56	71.38	8.06

**Table 2 materials-15-08848-t002:** The orientation index of crystal orientation with various annealing temperatures.

Annealing Temperature (°C)	P1	P2	P3	P4
460	1.303	1.606	1.074	1.14
500	1.243	1.804	1.309	1.158
540	1.084	2.129	1.266	0.976
580	1.063	1.496	1.195	0.885
620	0.837	1.498	1.039	0.728
660	1.072	2.075	1.456	0.767

**Table 3 materials-15-08848-t003:** Recrystallization frequency of Er metal with 60% cold-rolling deformation with various annealing durations.

**Annealing Duration (h)**	**0**	**0.5**	**1**	**1.5**	**2**	**2.5**
**Recrystallization frequency (%)**	6.6	86.9	95.3	89.0	96.4	95.8

## Data Availability

The data that supports the results of this study are available upon request from the authors.

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
