# Peer review of "The Influence of Annealing on the Microstructural and Textural Evolution of Cold-Rolled Er Metal"

_materials, 2022, doi:10.3390/ma15248848_

Round 1

Reviewer 1 Report

1. Page 2, lines 62-73. As I understood, final thickness of the rolled plate was 4,8 mm? Correct me if I am wrong. What was rolls rotational velocity while rolling? What was the length of the rolls? Where is the mill located? How many passes were in total? Was there some intermediate heat treatment between the passes?

2. Was the grain size estimated for the unrolled plate?

3. Is it possible to specify the rolling direction for Figure 2a? Is it perpendicular to the image plane?

4. Please, specify the rolling direction for the microstructure images of Figure 3, Figure 8 and Figure 12.

5.  In my version of the paper scale bars are not visible for all the images of Figure 8.

6.  Page 10, lines 219-232. You discuss uniformity of the grain sizes. Could you, please, describe it numerically?

Author Response

Dear reviewer:

Thank you for taking the time to review our paper “The influence of annealing on the microstructural and textural evolution of the cold rolling Er metal” (ID: materials-2046442). Those comments are all valuable and very helpful for revising and improving our paper, as well as the important guiding significance to our research. We have studied the comments carefully and have made the correction. Revised portions are shown in the word. 

Reviewer 2 Report

This research investigates the microstructural and textural evolution of 60% cold rolling deformation Erbium (Er) Metal (purity≥99.7%) during annealing. Electron backscattered diffraction (EBSD) and X-ray diffraction (XRD) tests were performed. The research results showed that the texture of the (0001) plane orientation strengthened, but there was no apparent enhancement of (01-10) and (-12-10) plane orientation with increasing annealing temperature. The recrystallization frequency and grain sizes gradually stabilized after the annealing duration of more than one h at 740°C; the annealing duration and the recrystallization frequency fitted to the equation: y=1-exp(-0.3269x0.2506). 

However the present work is interesting, but it also needs some revisions as follows:

1- What are the novelties of the present article?

2- Careful proofreading is needed before its publication in the journal. 

3- Some references were missed from the introduction section, such as the following articles:

https://doi.org/10.1007/s11106-018-9962-4

https://doi.org/10.3390/ma15144732

https://doi.org/10.3390/ma14226910

4- The quality of the figures is good.

5- What are the applications of cold-rolled Er sheets in the industry?

5- What did you conclude about the grain size distribution of Er metal with 60% cold rolling deformation with various annealing duration?

Author Response

(The authors gave the same response as above.)
